# Mindfulness and Regulatory Emotional Self-Efficacy of Injured Athletes Returning to Sports: The Mediating Role of Competitive State Anxiety and Athlete Burnout

**DOI:** 10.3390/ijerph191811702

**Published:** 2022-09-16

**Authors:** Yiwei Tang, Yang Liu, Longjun Jing, Huilin Wang, Jingyu Yang

**Affiliations:** 1School of Physical Education, Hunan University of Science and Technology, Xiangtan 411201, China; 2School of Business, Hunan University of Science and Technology, Xiangtan 411201, China; 3China Athletics College, Beijing Sport University, Beijing 100084, China; 4Faculty of Economics, Chulalongkorn University, Bangkok 10330, Thailand; 5Department of Medical Bioinformatics, University of Göttingen, 37077 Göttingen, Germany

**Keywords:** mindfulness, regulatory emotional self-efficacy, competitive state anxiety, athlete burnout

## Abstract

Usually, both external environmental factors and internal psychological factors affect the self-efficacy of athletes returning to sports after an injury. Based upon COR theory, this study investigated mindfulness interventions’ effects on competitive state anxiety and burnout in injured athletes who are returning to sports. The study was conducted in South China from March to April 2022. The snowball and convenience sampling methods were used to select high-level sports teams’ injured athletes returning to sports, and a questionnaire survey was administered, from which 433 valid samples were obtained. Amos v. 26 was used to analyze the data. The results showed that mindfulness has a significant negative effect on competitive state anxiety and burnout, such that after strengthening the mindfulness intervention, athletes’ competitive state anxiety and burnout decreased and regulatory emotional self-efficacy increased. Further, this study indicated that athletes are prone to negative emotions after injury, and among athletes who returned to sports after injury, those with mindfulness interventions reported lower levels of competitive state anxiety and burnout. Hence, the study demonstrated that mindfulness can improve regulatory emotional self-efficacy in injured athletes who are returning to sports by reducing competitive state anxiety and burnout.

## 1. Introduction

To pursue the peak sports performance of “faster, higher, and stronger”, athletes undergo arduous training constantly for a long period of time to exceed their limits [1], and thus, sports injuries in athletes are extremely common during competition and training. According to Kraus and Conroy’s [2] research, 3–5 million people experience different sports and leisure injuries every year. Because an injured athlete experiences both physical and psychological effects throughout the sports injury and recovery process, it is also regarded as a process that is accompanied by tension, depression, and other difficulties. Relevant studies believe that a sports injury is a major accident for athletes because it poses a significant threat to the injured person’s physical, emotional, and social life [3]. Sports injuries can have a range of adverse consequences for athletes, including physical pain, loss, disappointment, lack of purpose in life and confidence in the future, and isolation from one’s sport. Therefore, this explains generally why injured athletes experience emotional problems in competitive sports settings.

In the past ten years, sports injuries’ effects on the injured person’s mental health and their mental state’s effect on the recovery from physical injuries have been a concern. Given this, the phenomenon of sports injuries in athletes should not be underestimated. Studies have shown that injured athletes’ responses to sports injuries can be manifested in three aspects: cognition; emotion, and psychology. Although there are individual differences in these psychological responses’ intensity and duration, researchers have found that athletes often experience certain forms of cognitive and emotional responses from the onset of the injury to the recovery and return to training, which is manifested prominently in a range of emotional changes, including worry, depression, and frustration [4]. First, according to statistics such as those in Udry et al. [5], 24% of injured athletes reported that their first reaction after injury was pain. Second, an athlete’s cognitive response to injury is usually accompanied by a series of emotional responses. Severe sports injuries lead to intense mood swings and tension in athletes [6]. In another study, Leddy et al. [7] found that high-level athletes showed the same emotional changes and experienced greater depression and anxiety and lower self-esteem. Clearly, athletes are not immune to emotional disturbances because of their athletic ability and physical fitness. Further, as Wiese-Bjornstal et al. pointed out [8], a sports injury is a highly depressing event that elicits complex psychological responses related to the injury and recovery. In general, a combination of personal and environmental factors affect the elicitation of these psychological reactions, and it is more likely to lead athletes to experience post-injury anxiety and burnout when facing sports.

A review of previous studies showed that many scholars’ research on the treatment of injured athletes has focused primarily on the effect of social support on their return to sports, including support from coaches, teammates’ families, and professional medical teams [9,10,11]. In addition to external factors, such as social and environmental factors, there are also certain personal factors, such as individual personality. Therefore, this study attempted to achieve four objectives: (1) understand the effect of mindfulness interventions on injured athletes’ negative emotions when they return to sports; (2) explore the psychological factors that affect injured athletes when they return to sports; (3) investigate the mediating effect of competitive state anxiety and burnout based upon conservation of resource theory (COR) and further explore the influence and mechanism of mindfulness on regulatory emotional self-efficacy, and (4) make suggestions to address injured athletes’ psychological problems.

Given competitive state anxiety and burnout’s adverse effects on athletes, it is critical to understand their regulatory emotional self-efficacy during exercise. Compared with the existing research, the contribution of this study is reflected primarily in the following three aspects. First, athletes’ regulatory emotional self-efficacy after an injury is taken as the main research focus. Second, this study attempts to use athletes’ specific comprehensive physiological and psychological responses when they return to sports after an injury, i.e., competitive state anxiety and burnout, as mediating variables that influence their self-confidence in regulating emotions. Third, from the perspective of COR theory, when an individual loses certain specific resources, s/he experiences a combined physiological and psychological reaction, burnout. Burnout in athletes is identical to that in other individuals who become exhausted and overwhelmed, physically and/or emotionally, and lose their motivation to work. When athletes lose specific resources after injuries, such as psychological, social, physical, rehabilitation, and career concerns, anxiety can arise when participating in sports [12]. Therefore, this study uses the two variables of competitive state anxiety and burnout as mediating variables and uses mindfulness as an individual resource that influences competitive state anxiety and burnout. Further, this study analyzes mindfulness’s effect on regulatory emotional self-efficacy after injury, which improves the injured athletes’ control and self-confidence in regulating emotion and provides a theoretical and practical basis for athletes who are returning to sports after injury. In summary, the results of this study provide further support for previous studies and offer a reference for similar studies in the future.

The remainder of the study is organized as follows: Section 2 provides a literature review of the relevant theories and proposes hypotheses and conceptual models. Section 3 presents the data collection methods, questionnaire composition, and data analysis methods. Section 4 describes the results of the data analysis and the hypothesis tests. Section 5 discusses the results and presents practical implications. Section 6 summarizes the central idea of this study and provides recommendations for future research.

## 2. Literature Review

### Conservation of Resource Theory (COR Theory)

Hobfoll [13] proposed the conservation of resource theory in 1988. The COR theory is often used to explain the way people generate pressure attributable to the adjustment relation between environmental demand and resource supply and demand in general pressure. Its core idea is that individuals always attempt to seek and possess resources. When they perceive that existing resources may be lost or have been lost and expected resources cannot be obtained, it leads to psychological pressure and tension [12]. COR theory holds that when individuals lose specific resources or precious resources are threatened or insufficient to meet demand they will experience job burnout when they invest those resources but fail to receive the expected return. Among them, “precious resources” are defined as objects, abilities, time, and energy that are related closely to individuals. The source of stress in the process of acquiring other precious resources is also the principal cause of burnout [12,14].

According to COR theory, when some potential or actual resource is lost, it can lead to a negative state, such as depression, anxiety, and physical tension [13]. Under the influence of pressure from various factors, athletes experience such symptoms as depression and physical fatigue during normal training and competition. Burnout is a common stressor for athletes that depletes and threatens their critical resources. A small number of athletes (1–2%) will experience a more severe and chronic state of exhaustion that may eventually lead to complete withdrawal from the sport, a potential tragedy for the athlete [15].

In addition, COR theory states that individuals can cope with the depletion of their resources by accumulating more [16]. Personal resources, particularly those related to personal characteristics, can improve the individual’s ability to resist pressure and offset the damage to other personal resources [17]. An athlete’s characteristics can be considered resources, as they influence the way s/he responds to competition stress and other resource losses [18]. Therefore, it can be said that the athlete’s more personal resources may be able to offset the loss of other resources.

While COR theory primarily describes the nature of psychological stress and its consequences, the area of psychological burnout in high-level athletes has not been explored. This study focused on investigating the explanatory power of COR theory in the context of professional sports athletes. Relevant studies have shown that mindfulness plays an important role in relieving personal anxiety and depression and promoting a person’s positive emotional state [19]. From the perspective of COR theory, this study attempted to demonstrate that mindfulness can be used as a personal resource to relieve competitive state anxiety and burnout and improve regulatory emotional self-efficacy. Accordingly, this study explores new moderators and independent variables to determine whether they mediate the relation between burnout and emotion by using athletes’ post-injury psychological burnout as a mediator.

## 3. Hypotheses

### 3.1. Mindfulness, Competitive State Anxiety, and Athlete Burnout

From the perspective of COR theory, mindfulness can reduce emotional burnout and anxiety by relieving the adverse emotions athletes experience after an injury, and thus, it can be considered a personal trait of athletes after injury. Charoensukmongkol [20] asserted that mindfulness is the key “psychological resource” that optimizes individual psychology. Liu et al. [21] cited the COR theory to show that employees’ mindfulness is a special internal resource of the individual that helps enhance the positive psychological resources in the work environment.

Empirical studies have demonstrated that mindfulness leads to a variety of positive psychological effects, including increased subjective wellbeing, reduced adverse psychological symptoms and emotional responses, and improved behavioral regulation [22]. Mindfulness-based stress reduction can decrease state anxiety as well as enhance empathy and self-compassion [23]. Further, mindfulness-based cognitive therapy can be an effective antidepressant and anxiolytic intervention for athletes [24]. For example, in elite martial arts athletes, a mindfulness intervention may be associated with reduced anxiety about competitive states [25].

Further, Gustafsson et al. [26] explored whether perceived stress and different emotions mediate the relation between mindfulness and burnout in elite youth athletes, and the results suggested that emotions moderate the relation between mindfulness and physical burnout to a certain extent. For example, Jouper and Gustafsson [27] conducted a case study of elite athletes in marksmanship and showed that a mindfulness intervention may help them prevent and/or recover from burnout. In addition, Li et al. [28] demonstrated through controlled experiments and interview studies that there is a negative correlation between mindfulness and burnout.

In summary, mindfulness, competitive state anxiety, and burnout may be highly correlated theoretically, and from the perspective of COR theory, being mindful allows individuals to acquire additional important psychological resources. Mindfulness, then, as a positive psychological resource, can help athletes cope effectively with anxiety and burnout in competitive situations and offset the loss of other resources that athletes experience after injury. This reduces their negative psychological symptoms when they return to the field. On the one hand, previous studies have shown that mindfulness can effectively reduce anxiety, which indicates that there may be a negative correlation between mindfulness and competitive state anxiety. On the other hand, according to empirical evidence, negative emotions are among the factors that affect physical burnout, and a mindfulness intervention has a good regulating effect on negative emotions [26]. Therefore, it is conceivable that there may be a negative correlation between mindfulness and burnout. Thus, this study proposes Hypotheses 1 and 2:

**Hypothesis** **1** **(H1).**
*Mindfulness has a negative impact on competitive state anxiety.*


**Hypothesis** **2** **(H2).**
*Mindfulness has a negative impact on athlete burnout.*


### 3.2. Burnout and Regulatory Emotional Self-Efficacy

According to Bandura’s self-efficacy theory [29], as the core variable in an individual’s self-belief system, self-efficacy is an important resource and approach for individuals to cope with trauma. Regulatory emotional self-efficacy is a form of self-efficacy with emotion regulation characteristics, and its formation and negative emotions also affect its development [30]. Regulatory emotional self-efficacy refers to individuals’ confidence in their ability to regulate their emotions, which affects their success in doing so and is the basis of emotional competence [30].

Much research now focuses on improving regulatory emotional self-efficacy to relieve tension, maintain emotional regulation, help regulate emotional impulses, and promote mental health [31]. Regulatory emotional self-efficacy has a complex internal structure that includes negative emotion and positive emotion self-efficacy [32]. Researchers have also conducted empirical research on regulatory emotional self-efficacy’s mediating effect from different perspectives.

The importance of emotion regulation has been increasingly recognized in recent years, as it is associated with various forms of psychopathology, including anxiety [33], and it has also been shown to be correlated with anxiety and burnout. For example, Mathews et al.’s [34] research found a significant effect in the association between anxiety and emotional capacity in childhood and adolescence, which informs intervention and treatment plans for anxiety in adolescents. Similarly, Kashdan and Breen [35] studied social anxiety as a predictor of positive emotions and found that people with excessive social anxiety demonstrated low levels of positive emotions, which, when enhanced, will reduce social anxiety through emotion regulation measures. However, individual differences also affect emotion regulation. Further, studies have shown that, with respect to negative emotions, the mediating role regulatory emotional self-efficacy plays between negative emotions and behaviors has been partially confirmed [36]. During the COVID-19 epidemic, Chinese nurses were under great psychological pressure. A questionnaire survey was conducted with 339 nurses who were caring for COVID-19 patients, and the results showed that regulatory emotional self-efficacy had a significant relation with personality and negative emotions [37]. Thus, the studies above also verified that there is a correlation between regulatory emotional self-efficacy and anxiety. According to the COR theory, athletes are more vulnerable to the loss of personal resources post-injury because they must bear its physical, psychological, and social consequences. Athletes who are uninjured may be less vulnerable to resource loss because they face fewer hardships. Therefore, when athletes lose fewer resources, it has a positive effect on their regulatory emotional self-efficacy.

Burnout is a psychological syndrome characterized by emotional exhaustion, feelings of cynicism, and reduced personal fulfillment [38]. Research in various occupational domains has shown that regulatory emotional self-efficacy has an effect not only on anxiety but also on burnout. For example, Klusmann et al. [39] found that new teachers experienced job burnout in their first year of teaching practice, while experienced teachers experienced similar burnout but had higher job satisfaction. Emotional self-regulation can help alleviate burnout in teaching work and improve job satisfaction. Alessandri et al. [40] found that self-efficacy in managing negative emotions at work was significantly correlated with emotional stability and job burnout. Job burnout and athlete burnout may also have certain similarities. It can be seen as well that from the perspective of COR theory there is a correlation between athletes’ burnout and regulatory emotional self-efficacy.

In summary, various studies have shown that regulatory emotional self-efficacy is significantly related to anxiety and burnout, so competitive state anxiety, burnout, and regulatory emotional self-efficacy are likely highly correlated in theory. From the perspective of COR theory, both athletes’ competitive state anxiety and burnout are attributable to their losses of personal resources. On the one hand, an athlete’s anxiety about returning to sports appears to be influenced by the loss of personal resources following the injury, in which less loss of personal resources is likely to imply a lower competitive state anxiety, and lower competitive state anxiety also appears to predict higher emotional self-efficacy; on the other hand, according to COR theory, burnout is a form of emotional exhaustion that may affect an athlete’s emotional regulation self-efficacy. In this case, it is reasonable to assume that lower levels of burnout predict higher regulatory emotional self-efficacy. Based upon this, this study proposes the following hypotheses:

**Hypothesis** **3** **(H3).**
*Competitive state anxiety has a negative effect on regulatory emotional self-efficacy.*


**Hypothesis** **4** **(H4).**
*Burnout has a negative effect on regulatory emotional self-efficacy.*


### 3.3. The Mediating Roles of Competitive State Anxiety and Athlete Burnout

COR theory [12] states that an individual possesses limited personal resources (i.e., time, physical state, emotional state, attention, etc.), and those resources also affect each other. Individuals try their best to acquire, conserve, and maintain their limited resources, which then has an effect on individual elements’ formation and development. In particular, the loss of resources leads to negative emotions and stress.

Numerous studies have shown that mindful individuals are able to cope more effectively with stress. For example, Charoensukmongkol and Suthatorn [41] argued that mindful employees tend to exhibit high levels of optimism, adaptability, and self-efficacy, all of which play important roles in reducing stress. Beginning from the COR theory, after recovery, athletes’ stress attributable to the loss of personal resources can be classified into competitive state anxiety and burnout. Therefore, Chen et al. [42] argued that mindfulness training not only enhances athletes’ mental health and flow status and reduces competitive anxiety but also contributes to their athletic performance. In addition, Li et al.’s [28] meta-analysis showed an inverse relation between mindfulness and burnout. Therefore, competitive state anxiety appears to influence mindfulness’s effects on burnout.

Many scholars have studied and demonstrated the relation between mindfulness and regulatory emotional self-efficacy. Jin et al.’s [43] study found that regulatory emotional self-efficacy and subjective wellbeing played a partial mediating role between mindfulness and loneliness. This paper has previously demonstrated that there is a correlation between burnout and regulatory emotional self-efficacy. Based upon COR theory, if mindfulness affects regulatory emotional self-efficacy, then the latter may play an intermediary role in its relation to burnout.

In the field of sports science, burnout has become an increasingly common problem in competitive athletes, and given that it is the result of chronic stress and the work environment often triggers anxiety responses, there is a causal relation between burnout and anxiety [38]. Therefore, many researchers have conducted considerable research on anxiety and burnout in different fields [44]. Among them, Cho et al. [45] found that athletes experience multi-faceted stress in their sports career, which may lead to anxiety and burnout. Research suggests that there is an important and indirect pathway from controlling coaches to burnout through competitive state anxiety that affects trait anxiety and, in turn, burnout. Roy et al. [46] found that physician burnout is currently on the rise and demonstrated a correlation between anxiety and burnout among them. Further, they found that mindfulness training programs can effectively reduce anxiety. Therefore, as there is a correlation between anxiety and burnout, and mindfulness has an effect on burnout, it is assumed that there is a certain relation between these factors.

In summary, from the perspective of COR theory, first, relevant research has proven that mindfulness, as a personal psychological resource, has a positive effect on individuals’ burnout, and there is a mutual influence between the individuals’ resources. The former study demonstrated that there is a significant relation between competitive state anxiety and burnout. Thus, competitive state anxiety may partially mediate mindfulness‘s effect on burnout. Second, if there is a significant relation between burnout and regulatory emotional self-efficacy, does mindfulness have an effect on the latter through burnout? Finally, assuming that there is a relation between mindfulness, competitive state anxiety, burnout, and regulatory emotional self-efficacy, can mindfulness affect regulatory emotional self-efficacy through competitive state anxiety, and does burnout mediate regulatory emotional self-efficacy? Based upon this, this study proposes the following three hypotheses:

**Hypothesis** **5** **(H5).**
*Competitive state anxiety mediates the effect on the relation between mindfulness and burnout negatively.*


**Hypothesis** **6** **(H6).**
*Burnout mediates the effect on the relation between mindfulness and regulatory emotional self-efficacy negatively.*


**Hypothesis** **7** **(H7).**
*Competitive state anxiety and burnout have a negative mediating effect on the relation between mindfulness and regulatory emotional self-efficacy.*


A summary of all hypotheses is shown in Figure 1.

## 4. Materials and Methods

### 4.1. Participants and Procedure

This study examined high-level athletes (i.e., athletes at national level two and above) after injury and adopted the snowball and convenience sampling methods. The researchers conducted a questionnaire survey on high-level sports players in South China from April to May 2022. First, they contacted the leaders of high-level sports teams in each province and asked them to distribute questionnaires to the high-level sports teams in the provinces; further, the team members also shared the questionnaires with each other. Six hundred questionnaires were distributed, and by the end of May, a total of 485 high-level athletes had been surveyed. After invalid questionnaires (e.g., missing information and blank answers) were deleted, 433 valid questionnaires were obtained, for a response rate of 72.2%.

Table 1 lists the demographic characteristics of the surveyed athletes. Nearly 70% of the respondents were between the ages of 18 and 23; 61.9% were men and 38.1% were women; 77.4% were second-level athletes, and ball sports were the main sports category (62.8%).

### 4.2. Measures

To measure the effectiveness of mindfulness, the study used a scale Carlson and Brown [47] developed. The measure of the athletes’ competitive state anxiety was obtained from Martinent and Ferrand [48]. Further, burnout was measured using the five-item scale Raedeke and Smith developed [49], and the athletes’ regulatory emotional self-efficacy was measured using a scale Caprara et al. developed [32]. The four scales use a five-point Likert scale ranging from 1 (i.e., strongly disagree) to 5 (i.e., strongly agree).

To adapt to the research field and the specific cultural background, the researchers made certain adjustments to the scales’ items. To ensure the adjusted test scales’ reliability, a pilot test was conducted with high-level college athletes at a university in Changsha City [50]. The researchers distributed 70 questionnaires using the convenience sampling method and recovered 66 valid questionnaires. The results showed that the Cronbach’s alpha coefficients were all greater than 0.9, indicating that the scales had good internal consistency [51].

### 4.3. Data Analysis

Structural equation modeling (SEM) in AMOS v. 26.0 (with maximum-likelihood estimation) was used to analyze the proposed model. SEM is often used to evaluate latent variables in measurement models and test hypotheses between latent variables in structural models [52]. This study adopted the two-step modeling approach Anderson and Gerbing [53] proposed, i.e., the measurement model and the structural model were evaluated using SEM. First, the researchers assessed the model’s validity, i.e., the measured and structural models, using SEM; then, the fit and path coefficients of the hypothetical model were measured.

## 5. Results

### 5.1. Measurement Model

Fornell and Larcker [51] suggested that reliability analysis should include Cronbach’s alpha and composite reliability (CR) coefficients that measure latent variables. The reliability tests summarized in Table 2 show that the variables had Cronbach’s alpha coefficients in the range of 0.88–0.92, well above the recommended value of 0.7. The CR coefficients of the variables ranged from 0.89–0.9294, much higher than the 0.7 value Joseph et al. recommended. [54]. Therefore, all variables’ reliability values were good. Convergent validity refers to how similar the measurement results are when different measurement methods are used to determine the same feature. It is usually measured by factor loadings and average variance extracted (AVE) [51]. The results in Table 3 show that all measured items’ factor loadings were in the range of 0.76–0.91, and all variables’ AVE values were in the range of 0.66–0.71, which is higher than the value of 0.5 that Fornell and Larcker recommended [51]. Therefore, all variables had high convergence validity. In addition, researchers typically verify the data’s discriminative validity by comparing the correlation coefficient of each variable with the square root of the mean. The results are shown in Table 3; all correlation coefficients were less than the square root of AVE. Therefore, the variables demonstrated good discriminant validity.

### 5.2. Common Method Variance

The common method variance (CMV) issues that may exist in behavioral research were examined as well. First, Harman’s univariate test results showed that the percent of variance extracted from the univariate tests was 49.78% (below the classical threshold of 50%), implying the absence of CMV [55]. Second, the study followed the method Lindell and Whitney [56] proposed for CFA single-factor and two-factor comparisons. The single-factor model’s Chi-square value was 2320.1 with 135 degrees of freedom, while the multi-factor model had a Chi-square value of 251.5 and 129 degrees of freedom. The ratio of the difference in the two models’ Chi-square values to the difference in the degrees of freedom is 344.8, and the two models’ Chi-square difference was very significant, proving that there is no CMV. Therefore, CMV correction was not required in this study.

### 5.3. Structural Path Model

As the structural model’s error and residual terms have no negative values, it indicates that the model overall does not violate the basic fitness test criteria. With respect to the values Hair and Black et al. suggested [52], the structural model showed a good fit to the data (χ^2^/df = 2.77, GFI = 0.92, NFI = 0.94, CFI = 0.98, TLI = 0.95, and RMSEA = 0.06). Table 3 lists the correlations among the variables and shows that there were significant correlations among the independent variables, mediating variables, and dependent variables, which provided initial support for the research hypotheses. The results of the structural pathway model are shown in Figure 2; mindfulness’s effect on competitive state anxiety was statistically significant (*β* = −0.58, *p* < 0.001), and supported H1, as was its effect on burnout (*β* = 0.51, *p* < 0.001), which supported H2. Competitive state anxiety had a statistically significant effect on regulatory emotional self-efficacy (*β* = −0.25, *p* < 0.01), supporting H3. Finally, burnout’s effect on regulatory emotional self-efficacy was statistically significant (*β* = −0.41, *p* < 0.001), which supported H4.

The conceptual models showed that mindfulness positively affects athletes’ regulatory emotional self-efficacy through the two proposed mediators, competitive state anxiety and burnout. This study followed Bollen and Stine’s [57] recommendation and used a bootstrapping method to verify the mediation effect. Table 4 presents the results of 5000 bootstrap samples with a 95% confidence interval; all Z values were greater than 1.96, and there were no zeros within the 95% confidence interval. In addition, the study showed that there was a significant mediating effect between mindfulness and burnout via competitive state anxiety (standardized indirect effect = −0.16, *p* < 0.001), which provided support for H5. There was a significant mediating effect between mindfulness and regulatory emotional self-efficacy through burnout (standardized indirect effect = 0.42, *p* < 0.001), which provided support for H6. The study also showed a significant mediating effect between mindfulness and regulatory emotional self-efficacy through competitive state anxiety and burnout (standardized indirect effect = 0.42, *p* < 0.001), which supported H7. The results of the study showed that athletes‘ competitive state anxiety and burnout decreased significantly after the mindfulness intervention, and thus, it was able to reduce their negative emotions, improve their positive emotions, and help athletes when they returned to sports.

## 6. Discussion

### 6.1. Theoretical Contribution

This study investigated a mindfulness intervention’s effects on injured athletes’ psychological factors (i.e., competitive state anxiety, burnout, and regulatory emotional self-efficacy) based upon COR theory. The results showed that mindfulness had a significant negative effect on competitive state anxiety, and competitive state anxiety and burnout mediated mindfulness’s effect on regulatory emotional self-efficacy. Mindfulness had the greatest effect on competitive state anxiety, followed by burnout. As shown in Figure 2, the extended COR model explained 34% of the variance in regulatory emotional self-efficacy, much higher than the approximately 20% variance in previous studies. Taken together, these results are consistent with previous research showing that mindfulness can reduce anxiety and burnout in the face of stress in athletes [25,27], especially those recovering from injury. Furthermore, mindfulness interventions had positive effects on mental health and the regulation of negative emotions in injured athletes, which is consistent with the study by Mohammed et al. [58]. The results showed that athletes’ mental problems can be alleviated through a method other than counseling and that mindfulness is a more effective way to do so.

This study provides a theoretical contribution to mindfulness research in injured athletes, and the role of mindfulness in reducing injured athletes’ regulatory emotional self-efficacy provides additional support for the COR theory. First, these findings provide additional evidence for the research by Charoensukmongkol [20], who argued that mindfulness is an important psychological resource for optimizing individuals. The findings are similar to those of Montani et al. [59] and Liu et al. [21], who used COR theory to explain that mindfulness functions as an individual-specific internal resource that helps individuals enhance positive psychological resources in the workplace. Second, this study expanded on previous models using COR theory to elucidate mediating conditions that are more explicit about the benefits of mindfulness to enhance regulatory emotional self-efficacy in injured athletes.

From a theoretical point of view, these findings provide additional evidence for the COR theory, which in part suggests that the role of mindfulness appears to be particularly important in injured athletes experiencing emotional anxiety and burnout [16]. In addition, the theoretical value of this study lies in analyzing the effect of mindfulness on regulatory emotional self-efficacy. According to COR theory, competitive state anxiety and burnout may be seen as sports characteristics that lead to the resource depletion experienced by injured athletes returning to sports. Because mindfulness, as a psychological resource, can make up for resources lost due to the competitive pressures of injured athletes returning to sports, this provides a theoretical explanation for the personal characteristics of injured athletes and could reinforce the importance of mindfulness in helping athletes cope with the stress of competitive sports. The results of this study help to expand the existing knowledge of mindfulness research in injured athletes, a group susceptible to negative emotions [60], but less experience has been gained so far. These findings add to the limited research on the effects of mindfulness on injured athletes returning to sports [58]. Most importantly, this study expanded on previous studies by Mohammed et al. [58] and Naderi et al. [61], showing that certain conditions leading to mindfulness are more or less helpful in enhancing regulatory emotional self-efficacy in injured athletes.

### 6.2. Practical Implications

This study suggests the use of interventions to help injured athletes reduce competition stress, which is closely related to the Chinese competitive sports industry. Considering the injuries that athletes inflict during training and competition, this inevitably leads to negative emotions in athletes [4]. Therefore, some suggestions are put forward in response to the research results. First, for the national sports bureaus, considering mindfulness’s positive effect on competitive state anxiety, burnout, and regulatory emotional self-efficacy, the sports industries in various countries should strengthen the degree of policy implementation and optimize training to promote the efficiency of mindfulness training. For countries with relatively rapid psychological development, the popularity and accuracy of mindfulness training will be greater, so the government can conduct academic exchanges with such countries to improve the level and professionalism of mindfulness training. At the same time, high-level sports teams can cooperate with universities or research institutes to build a mental training system for athletes, including mindfulness training, so that athletes can join future training and competitions as soon as possible after recovery.

In addition, the national sports bureaus should also increase the proportion of funds invested in mindfulness training in daily training. This begins by increasing the ownership of mental and motor assessment equipment. Mindfulness training technology must be mastered slowly, and related equipment is required to provide stable and accurate long-term evaluation data to monitor the athletes’ status and formulate targeted training plans. To improve the athletes’ and coaches’ understanding of their sports state further, it can be used to improve the more controllable training factors, such as professional skills, strength, flexibility, and recovery training, to improve athletes’ regulatory emotional self-efficacy. In addition, sports departments at all levels should further enrich mindfulness training resources, actively promote the construction of athletes’ mental health, and improve athletes’ awareness of the importance of mindfulness training.

Second, for coaches, the coaches’ level of understanding of psychological principles is an important factor in improving athletes’ mindfulness. Therefore, the general administrations of sport of various countries need to add more psychological training skills courses to coach training to improve their level of mindfulness training. Beginning with the coach, we can improve the athletes’ level of mindfulness learning and training efficiency, obtain timely feedback on the association between their personal exercise state and psychological state, promote the improvement of training, and reduce athletes’ competitive state anxiety and burnout. This will offer a greater ability to continue to achieve regulatory emotional self-efficiency.

Third, there are important practical implications for athletes to providing mindfulness interventions to reduce the deleterious effects of sports injuries on the emotional and mental health of athletes. Given that the findings confirmed a positive correlation between mindfulness and regulatory emotional self-efficacy. Coaches need guidance on mindfulness training for injured athletes to help them develop and improve the quality of mindfulness training and allow athletes to engage in mindfulness practices, such as meditation, during breaks or before training games to help them relax from the stress of training and competition. Research by Chen et al. [42] supports the idea of mindfulness training, suggesting that athletes need mindfulness interventions to improve their performance on the field. Therefore, mindfulness training may also be an effective intervention for injured athletes to help improve their regulatory emotional self-efficacy when they return to sports.

### 6.3. Limitations

This study has certain limitations. First, non-random sampling was used, which helps researchers collect a large amount of data in a short period, but the sample’s representativeness may be biased. In future research, researchers should use a random sampling method so that the obtained samples will be more evenly distributed in the population, which is conducive to improving sample representativeness, reducing sampling error, and improving the accuracy of sample results. Second, the mindfulness intervention considered here had only a relatively shallow level. Future research should consider more aspects of mindfulness therapy, including mindfulness-based stress reduction (MBSR), mindfulness-based cognitive therapy (MBCT), and dialectical behavioral therapy (DBT), among others. Third, this study did not provide an alternative model, and future research can provide more possibilities on this basis. Further, the sample size in this study can be increased in future work to realize the general application of the research results.

## 7. Conclusions

In response to the proposed research objectives, this study demonstrated that the vast majority of high-level post-injury athletes were able to overcome competitive state anxiety and burnout in the face of returning to sports under the influence of a mindfulness intervention. At present, mindfulness interventions are quite effective in helping athletes regulate their emotions. Particularly after an injury, it is important for athletes to maintain a positive attitude toward sports, adjust their self-efficacy, and enhance their confidence and willpower. Further, the findings suggest that mindfulness is an important factor that influences athletes’ competitive state anxiety and burnout after injury. Mindfulness was found to affect athletes’ regulatory emotional self-efficacy directly or through the mediating effects of competitive state anxiety and burnout. Therefore, this study suggests that, post-injury, athletes should engage in mindfulness training during exercise to effectively reduce competitive state anxiety and burnout. Coaches and athletes should pay attention not only to their mental health but also learn to control their emotions and regulate their competitive state.

## Figures and Tables

**Figure 1 ijerph-19-11702-f001:**
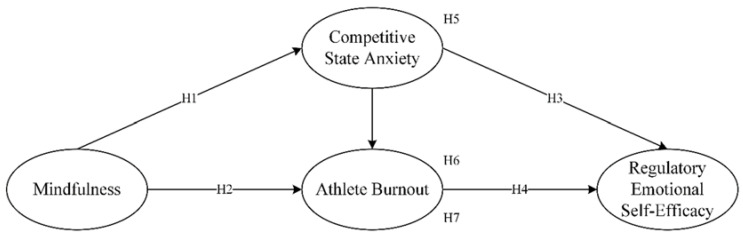
The hypothesized model.

**Figure 2 ijerph-19-11702-f002:**
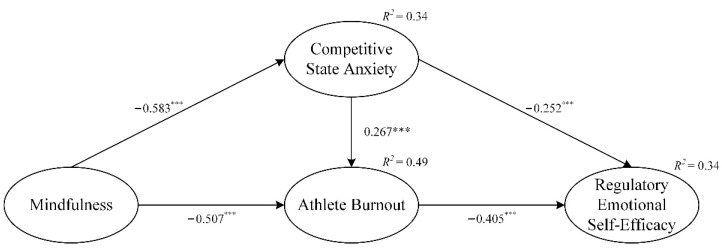
Structural path model. *** *p* < 0.001. Standardized coefficients are reported.

**Table 1 ijerph-19-11702-t001:** Participant profile (N = 433).

Title	Profiles	Survey (%)
Respondent age	≤17	28 (6.5%)
	18–20	121 (27.9%)
	21–23	182 (42%)
	≥24	102 (23.6%)
Respondent gender	Male	268 (61.9%)
	Female	165 (38.1%)
Respondent sport level	Second-level athlete	335 (77.4%)
	Tier 1 athlete	82 (18.9%)
	Athletes at the gym level	16 (3.7%)
Respondent sports items	Ball sports	272 (62.8%)
	Athletics projects	108 (24.9%)
	Other sports	53 (12.3%)

**Table 2 ijerph-19-11702-t002:** Reliability and validity tests.

Dimensions	Loadings	Cα	AVE	CR
** *Regulatory Emotional Self-Efficacy (RES)* **		0.905	0.705	0.905
RES1: You express joy when good things happen to you.	0.819			
RES2: You feel gratified by achieving what you set out to do.	0.836			
RES3: You avoid getting upset when others keep giving you a hard time.	0.845			
RES4: You reduce your distress when you do not receive the appreciation you feel you deserve.	0.858			
** *Competitive State Anxiety (CSA)* **		0.922	0.709	0.924
CSA1: I feel self-confidence.	0.799			
CSA2: I am worried about choking under pressure.	0.789			
CSA3: I am worried that I may not do well in this competition.	0.894			
CSA4: I am confident because I mentally picture myself reaching my goal.	0.907			
CSA5: I am worried that others will be disappointed with my performance.	0.813			
** *Athlete Burnout (ABS)* **		0.921	0.703	0.922
ABS1: I feel so tired from the training that I do not find the energy to do other things.	0.783			
ABS2: I am not as interested in the sport as I used to be.	0.809			
ABS3: I am exhausted by the physical and mental demands of the sport.	0.837			
ABS4: No matter what I do in sport, I do not perform as well as I should.	0.866			
ABS5: I have negative feelings towards the sport.	0.892			
** *Mindfulness (MIN)* **		0.883	0.658	0.885
MAAS2: I break or spill things because of carelessness, not paying attention, or thinking of something else.	0.815			
MAAS5: It seems I am “running on automatic”, without much awareness of what I am doing.	0.840			
MAAS6: I rush through activities without really being attentive to them.	0.825			
MAAS7: I get so focused on the goal I want to achieve that I lose touch with what I am doing right now to get there.	0.763			

All standardized loadings are significant at the 0.001 level.

**Table 3 ijerph-19-11702-t003:** Discriminant validity test.

Construct	ERS	CSA	ABS	MIN
RES	**(0.840)**			
CSA	−0.433 **	**(0.842)**		
ABS	−0.487 **	0.535 **	**(0.838)**	
MIN	0.661 **	−0.537 **	−0.592 **	**(0.811)**

The square root of the average variance extracted (AVE) is in diagonals (bold); off diagonals are Pearson’s correlations of constructs. ** *p* < 0.01.

**Table 4 ijerph-19-11702-t004:** Standardized direct, indirect, and total effects.

	Point Estimate	Product of Coefficients		Bootstrapping				
				95% CI		Bias-Corrected 95% CI		Two-Tailed Significance
		*SE*	*Z*	Lower	Upper	Lower	Upper	
** *Standardized direct effects* **								
MIN→CSA	−0.583	0.054	−10.796	−0.685	−0.473	−0.684	−0.472	0.000 (***)
MIN→ABS	−0.507	0.069	−7.348	−0.637	−0.368	−0.638	−0.369	0.000 (***)
CSA→ABS	0.267	0.066	4.045	0.140	0.392	0.141	0.393	0.000 (***)
CSA→RES	−0.252	0.074	−3.405	−0.398	−0.109	−0.397	−0.107	0.002 (**)
ABS→RES	−0.405	0.067	−6.045	−0.532	−0.265	−0.532	−0.265	0.000 (***)
** *Standardized indirect effects* **								
MAS→ABS	−0.156	0.040	−3.900	−0.233	−0.081	−0.236	−0.083	0.000 (***)
MAS→RES	0.415	0.052	7.981	0.312	0.516	0.309	0.513	0.001 (**)
CSA→RES	−0.108	0.028	−3.857	−0.164	−0.055	−0.171	−0.061	0.000 (***)
** *Standardized total effects* **								
MAS→CSA	−0.583	0.054	−10.796	−0.685	−0.473	−0.684	−0.472	0.000 (***)
MAS→ABS	−0.663	0.045	−14.733	−0.743	−0.564	−0.744	−0.566	0.000 (***)
MAS→RES	0.415	0.052	7.981	0.312	0.516	0.309	0.513	0.001 (**)
CSA→ABS	0.267	0.066	4.045	0.140	0.392	0.141	0.393	0.000 (***)
CSA→RES	−0.306	0.064	−4.781	−0.480	−0.235	−0.483	−0.237	0.000 (***)
ABS→RES	−0.405	0.067	−6.045	−0.532	−0.265	−0.532	−0.265	0.000 (***)

Standardized estimation of 5000 bootstrap samples; ** *p* < 0.01, *** *p* < 0.001.

## Data Availability

Not applicable.

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
