# Peer review of "Mindfulness and Regulatory Emotional Self-Efficacy of Injured Athletes Returning to Sports: The Mediating Role of Competitive State Anxiety and Athlete Burnout"

_ijerph, 2022, doi:10.3390/ijerph191811702_

Round 1

Reviewer 1 Report

Mindfulness and regulatory emotional self-efficacy of injured 2 athletes returning to sports is an interesting topic. In this paper, the snowball method and convenient sampling method are used to conduct a questionnaire survey on the injured athletes of high-level sports teams who return to the competition.. The research methods are properly used and good results are obtained. Overall, the quality of the paper is good, and I think it meets the requirements for publication. I agree to publish it after a minor revision.

The shortcomings of the paper are as follows: First, the theoretical contribution (6.1) in the discussion section is too simple. I think it is necessary to expand the discussion and carry out a full discussion or explain your specific theoretical contribution, especially the comparison with previous studies and the important contributions in what aspects. Second, in the section 6.2. Practical implications, some good management recommendations were made for the General Administration of Sport, but was the General Administration of Sport the only one who needed to be concerned with our recommendations? Therefore, I suggest that the paper expand the scope of policy recommendations. Third, while describing the first limitation, the paper points out the limitation of non-random sampling, but does not explain the direction of future improvement. Please supplement it.

Reviewer 2 Report

Dear, the present study aimed to investigate the effects of mindfulness interventions on competitive state anxiety and burnout in injured athletes who returned to sport. 

The text presents a clear and in-depth exposition of the theoretical framework, followed by an argumentation of the justifications for carrying out the study, concluding with clear and direct hypotheses. In general, the methodological procedures are well structured and accurately described.

A strong point of the study is the statistical procedures and their results. I believe that the findings can shed light on future researchers. Finally, time managed to discuss the results in an academic way, associating practical knowledge.

Reviewer 3 Report

The article "Mindful and regulation emotional self-efficacy of injured athletes returning to sports. the mediating role of competitive state anxiety and athletes burnout" is interesting and well written. However, some aspects have to be improved.

 I thnk that the major problems concern how an injuries is considered and treated. First of all,  what do you mean as "injury"? any event that prevents the person from training? The injury is a complex event, are there any differences between training and race injuries? Are there differences between the various types of injuries? I think there are big differences in dealing with a ruptured anterior cruciate compared to dealing with a sprained ankle. Recovery times are also significantly different. I think there are also significant differences depending on the sport that is being considered.  Moreover, even the age at which you are injured can affect the perception and recovery from the injury. So I wonder if it is not a good idea to talk in general about injuries without considering all these aspects. 

In the introduction, I think it is important to specify if the data are about the general population or the athletes. Lines 34-35, a reference is needed.

Chapter 2, is too long and I think it is not a "literature review", but only a list of hypotheses related to a different theory. I think that this part could confuse the readers. 

Did you do a power analysis to understand the size of the sample? There are no indications about the sports considered. In addition, it is not clear if the athletes that answered the questionnaire were injured or if they had to think about their last injuries. And What were the possible answers to the questions?

Line 419 could be 5.1, not 6.1

Discussion must be improved: they are too short and there is no comparison with previous literature.

The limitations could be moved to the end of the discussion part, instead of conclusions.

Round 2

Reviewer 3 Report

I thank the authors for following the suggestions and improving the article, now, in my opinion, is ready for publication.